# SciRepEval: A Multi-Format Benchmark for Scientific Document Representations

**Amanpreet Singh**[1]      **Mike D'Arcy**[2]      **Arman Cohan**[1,3]
**Doug Downey**[1,2]      **Sergey Feldman**[1]
[1] Allen Institute for Artificial Intelligence, Seattle, WA, USA
[2] Northwestern University, IL, USA
[3] Yale University, CT, USA
{amanpreets, miked, armanc, dougd, sergey}@allenai.org

## Abstract

Learned representations of scientific documents can serve as valuable input features for downstream tasks without further fine-tuning. However, existing benchmarks for evaluating these representations fail to capture the diversity of relevant tasks. In response, we introduce SciRepEval, the first comprehensive benchmark for training and evaluating scientific document representations. It includes 24 challenging and realistic tasks, 8 of which are new, across four formats: classification, regression, ranking and search. We then use this benchmark to study and improve the generalization ability of scientific document representation models. We show how state-of-the-art models like SPECTER and SciNCL struggle to generalize across the task formats, and that simple multi-task training fails to improve them. However, a new approach that learns multiple embeddings per document, each tailored to a different format, can improve performance. We experiment with task-format-specific control codes and adapters and find they outperform the existing single-embedding state-of-the-art by over 2 points absolute. We release the resulting family of multi-format models, called SPECTER2, for the community to use and build on.

## 1 Introduction

Learning representations of documents is critical for a variety of NLP tasks such as classification, search, and recommendation (Cohan et al., 2020). Recent work has shown how pre-trained language models (e.g. (Devlin et al., 2019; Raffel et al., 2020; Brown et al., 2020)) can be tailored to produce high-quality document representations with contrastive learning (Xu et al., 2021; Gao et al., 2021; Neelakantan et al., 2022). In the scientific domain, contrastive learning of cross-document links (e.g. citations) has led to improved document-level representations (Cohan et al., 2020; Ostendorff et al., 2022b; Mysore et al., 2022). These representations

can be indexed and consumed later by lightweight downstream models without additional fine-tuning.

While there has been significant progress in evaluating generalizability of NLP models (Ye et al., 2021; Sanh et al., 2021), evaluation of scientific document representations has remained limited. Existing benchmarks either focus on document similarity (Mysore et al., 2021; Voorhees et al., 2021) or tasks that are highly correlated and not diverse (Cohan et al., 2020). Further, as shown in our experiments, a model's good performance on general-purpose text embedding benchmarks like the MTEB (Muennighoff et al., 2022) may not translate as well to scientific tasks.

We introduce SciRepEval, the first benchmark for comprehensive evaluation of document-representation models in the scientific domain. Unlike prior work, SciRepEval is large and includes a collection of highly diverse tasks, thus encouraging research on generalization (instance-level, cross-task and cross-domain). It consists of 24 realistic tasks that reflect practical use cases of scientific document representations across four formats: text classification, regression, proximity-based ranking (e.g., nearest-neighbor), and ad-hoc search. Eight of these are new contributions. SciRepEval contains standard sets of both training and evaluation datasets to simplify and standardize comparisons between methods evaluated on the benchmark.

Further, we use this benchmark to investigate and improve the generalization ability of document representation models. Following recent work (Cohan et al., 2020; Ostendorff et al., 2022b; Mysore et al., 2022) we pre-fine-tune a transformer model originally trained on citation triplets to produce high-quality representations for downstream tasks. We hypothesize that condensing all relevant information of a document into a single vector might not be expressive enough for generalizing across a wide range of tasks. Prior work addresses a similar challenge in the context of document similarity and

learns multiple representations associated with different *aspects* of a paper (e.g. task, method, results) (Mysore et al., 2022; Ostendorff et al., 2022a). In contrast, we aim to learn effective representations for multiple downstream task *formats*.

With the success of multi-task learning in NLP (Ye et al., 2021; Sanh et al., 2021), we explore it in the context of scientific document representations by optimizing a suitable objective for every task format in SciRepEval, i.e., cross-entropy for classification, triplet margin for proximity/ad-hoc search, and mean squared error for regression. We explore two state-of-the-art methods to generate format-specific representations: control codes (Keskar et al., 2019; Raffel et al., 2020) as input signal; and parameter-efficient adapter methods (Pfeiffer et al., 2021; Stickland and Murray, 2019), where a separate network module is introduced for every task format as in Figure 1.

Our experiments investigate (*i*) if existing document representation methods generalize to a highly diverse set of tasks, (*ii*) if multi-task training on diverse data can improve document representation models, and (*iii*) if task-format-specific representations can improve generalization. Through extensive analysis we find that existing state-of-the-art scientific document representation models such as SPECTER (Cohan et al., 2020) and SciNCL (Ostendorff et al., 2022b) struggle with generalizing to multiple task types. We interestingly find that simple multi-task training on a large set of tasks does *not* lead to significant improvements. However, we learn that multiple task format-specific representations can substantially improve generalization.

To summarize, our contributions are:

(i) SciRepEval, a new comprehensive benchmark of 24 highly diverse and practical tasks for scientific document representation techniques across four different formats, of which 8 are made available for the first time, and six are explicitly designed for training.

(ii) An extensive investigation on the generalizability of state-of-the-art scientific document representation models.

(iii) *SPECTER2*, a set of new multi-task document representation models that, unlike existing methods, can produce representations tailored to different task formats. The new methods show improved generalization, outperforming prior work by over 2 points absolute.

We release the benchmark and code to encourage further research in this area: `https://github.com/allenai/scirepeval`. The SPECTER2 models are released as well: `https://github.com/allenai/SPECTER2`.

## 2 Background

**Representing Scientific Documents**   Prior works have produced large scale language models pre-trained on scientific corpora (Beltagy et al., 2019; Yasunaga et al., 2022; Trewartha et al., 2022). These tend to perform better than general purpose models on scientific domain tasks, and serve as a foundation for learning dense embeddings of scientific documents. Cohan et al. (2020) and Ostendorff et al. (2022b) fine-tune SciBERT (Beltagy et al., 2019) with a triplet loss that encourages papers citing each other to have similar embeddings, using the title and abstract of research papers as the input.

Both Cohan et al. (2020) and Ostendorff et al. (2022b) are evaluated on SciDocs. However, as discussed in section 3 and Appendix G, this benchmark has important limitations. In contrast, SciRepEval provides more challenging and diverse tasks to help motivate methods for producing scientific document representations that can generalize well across tasks. We attempt to learn task-specific embeddings of documents by pre-fine-tuning on multiple objectives simultaneously. Ostendorff et al. (2022a) and Mysore et al. (2022) study the orthogonal task of generating multiple embeddings per paper for different "facets," while we aim to learn general embeddings for mutliple task formats.

**Multi-Task Learning Across Formats**   Multi-task learning (Caruana, 1993) with deep neural networks has shown improvements upon single-task training for related objectives (Liu et al., 2015, 2019b). Though unrelated tasks can lead to negative transfer, recent work shows that simply increasing number of tasks yields better performance in multi-task learning (Aghajanyan et al., 2021; Aribandi et al., 2022; Padmakumar et al., 2022). Aghajanyan et al. (2021) pre-fine-tune language models on 46 tasks from 4 task types before fine-tuning on the downstream task. Aribandi et al. (2022) pre-train T5 (Raffel et al., 2020) on C4 span denoising and 107 other tasks across 8 task families. Ye et al. (2021) introduce an ontology of 160 tasks for few shot multi-task training. Unlike these task families, which are divided primarily by semantics (e.g. classifying sentiment vs entailment), the training tasks in SciRepEval consist of 8 large scientific datasets across the four task *formats*. We wish to

evaluate final document representations, so rather than fine-tune on downstream tasks as in above approaches, we directly apply the representations as features to the tasks (Cohan et al., 2020).

**Adapters for Multiple Tasks** Adapters were introduced by Houlsby et al. (2019) for parameter efficient training of transformers (Vaswani et al., 2017). A small number of trainable parameters are added to each layer, while freezing the base encoder. This is similar to ELMo (Peters et al., 2018), which learned task-specific weightings for the biLSTM layers. To use adapters in a multi-task setup, Pfeiffer et al. (2021) define a two-step process they call Fusion. First, individual adapter modules are trained for every task. The second step adds fusion modules at each layer which attend to (i.e. fuse) all the previously pre-fine-tuned adapters, keeping them fixed. Similarly, Stickland and Murray (2019) introduced Projected Attention Layers (PALs) with adapters and self-attention modules for every task, but the entire network is trained simultaneously.

**Control Codes** Control codes can be defined as token(s) pre-pended to the input as additional signals for the model. Keskar et al. (2019) use control codes as prompts to govern style, content, and task-specific behavior for conditional text generation. Tay et al. (2022) use control codes to switch between three de-noising modes during pre-training, and associate every downstream task with a mode during fine-tuning. Zhang et al. (2022) apply control codes to dense retrieval and produce multiple representations covering different aspects of the same document, allowing them to match queries written from multiple perspectives. In contrast to this, we use control codes to indicate the target task format for an embedding output by the model, and demonstrate how this is effective for producing paper embeddings across different formats.

## 3 SciRepEval

We introduce SciRepEval, a benchmark of 24 tasks across four formats to train and evaluate multi-task embeddings of scholarly papers. SciRepEval aims to enable comprehensive evaluation of paper embeddings with: (1) a highly diverse set of tasks spanning multiple formats such as classification, regression, proximity and ad-hoc search to challenge the general-purpose applicability of embeddings, (2) realistic tasks that reflect practical use cases of paper embeddings, and (3) a standard set of training and evaluation datasets to simplify comparisons

between methods evaluated on the benchmark.

The previous scholarly paper embedding benchmark SciDocs (Cohan et al., 2020) includes two classification, one recommendation, and four nearest neighbors tasks. SciRepEval includes SciDocs as a subset, but addresses several key limitations:

**(i)** The four nearest neighbor tasks in SciDocs are built to distinguish related papers from random negatives given a query paper, which might be too easy and not representative of real tasks in scholarly information retrieval. SciRepEval has more realistic tasks like search, author disambiguation, and paper-reviewer matching among others.

**(ii)** For the methods evaluated in section 5, we found that the SciDocs recommendations task was noisy and had limited power to distinguish different embeddings. The test set includes only 1000 click-through events, and the use of propensity weighting means an even fewer examples dominate test performance. While SciRepEval includes SciDocs as a subset, we exclude the recommendations task.

**(iii)** The tasks in SciDocs were constructed to be used *only* for evaluation, and have few-enough samples that training on SciDocs is impractical (see Table 1). In SciRepEval, eight of the largest tasks across the four formats are intended for training, while the rest of the *out-of-train* tasks are reserved for evaluation. This enables the study of multi-task approaches, rather than relying solely on the citation signal. The training data in SciRepEval also has a large scale representation across multiple domains as discussed in Appendix E.

**(iv)** Four of the tasks in SciDocs have very high model-performance correlations among them (over 0.99), indicating that the diversity of the tasks is limited. See Appendix G for more details.

The tasks in SciRepEval are summarized in Table 1. They are a mixture of existing and new datasets. Datasets with at least 200,000 instances (triplets for proximity/ad-hoc search) are *in-train* datasets used for training while others are *out-of-train* used only for evaluation.

Next, we briefly describe each of the task formats and their component tasks. Full details are provided in Appendix A. All the tasks save one use paper embeddings created from a combination of paper title and abstract as the input. *Search* requires additional metadata (subsection 4.1) which is concatenated to the title and abstract.

**Ad-Hoc Search** In ad-hoc search tasks, we are given a short textual query and the aim is to rank a

| Task Format | Name | Train + Dev | Test | Eval Metric | Source |
|---|---|---|---|---|---|
| *In-Train* | | | | | |
| CLF | MeSH Descriptors | 2,328,179 | 258,687 | Macro F1 | **This work** |
| | Fields of study (FoS) | 676,524 **S** | 471 **G** | Macro F1 | **This work** |
| RGN | Citation count | 202,774 | 30,058 | Kendall's $\mathcal{T}$ | **This work** |
| | Year of Publication | 218,864 | 30,000 | Kendall's $\mathcal{T}$ | **This work** |
| PRX | Same Author Detection | **Q:** 76,489 **P:** 673,170 | **Q:** 13,585 **P:** 123,430 | MAP | (Subramanian et al., 2021) |
| | Highly Influential Citations | **Q:** 65,982 **P:** 2,004,688 | **Q:** 1,199 **P:** 58,255 | MAP | **This work** |
| | Citation Prediction Triplets | 819,836 | — | *not used for eval | (Cohan et al., 2020) |
| SRCH | Search | **Q:** 528,497 **P:** 5,284,970 | **Q:** 2,585 **P:** 25,850 | nDGC | **This work** |
| *Out-of-Train* | | | | | |
| CLF | Biomimicry | — | 10,991 | Binary F1 | (Shyam et al., 2019) |
| | DRSM | — | 7,520 **S**; 955 **G** | Macro F1 | (Burns, 2022) |
| | SciDocs MAG | — | 23,540 | Macro F1 | (Cohan et al., 2020) |
| | SciDocs MeSH Diseases | — | 25,003 | Macro F1 | (Cohan et al., 2020) |
| RGN | Peer Review Score | — | 10,210 | Kendall's $\mathcal{T}$ | **This work** |
| | h-Index of Authors | — | 8,438 | Kendall's $\mathcal{T}$ | **This work** |
| | Tweet Mentions | — | 25,655 | Kendall's $\mathcal{T}$ | (Jain and Singh, 2021) |
| PRX | S2AND | — | **X:** 68,968 **Y:** 10,942 | $B^3$ F1 | (Subramanian et al., 2021) |
| | Paper-Reviewer Matching | — | **Q:** 107 **P:** 1,729 | P@5, P@10 | (Mimno and McCallum, 2007) (Liu et al., 2014) (Zhao et al., 2022) |
| | RELISH | — | **Q:** 3190 **P:** 191,245 | nDCG | (Brown et al., 2019) |
| | SciDocs Co-view | — | **Q:** 1,000 **P:** 29,978 | MAP, nDCG | (Cohan et al., 2020) |
| | SciDocs Co-read | — | **Q:** 1,000 **P:** 29,977 | MAP, nDCG | (Cohan et al., 2020) |
| | SciDocs Cite | — | **Q:** 1,000 **P:** 29,928 | MAP, nDCG | (Cohan et al., 2020) |
| | SciDocs Co-cite | — | **Q:** 1,000 **P:** 29,949 | MAP, nDCG | (Cohan et al., 2020) |
| SRCH | NFCorpus | — | **Q:** 323 **P:** 44,634 | nDCG | (Boteva et al., 2016) |
| | TREC-CoVID | — | **Q:** 50 **P:** 69,318 | nDCG | (Voorhees et al., 2021) |

Table 1: Summary of SciRepEval tasks across the four formats - classification (CLF), regression (RGN), proximity (PRX) and adhoc search (SRCH). The models in section 6 are trained on the *in-train* tasks and then benchmarked on their held-out sets as well as the 16 test tasks. Information retrieval tasks have **Q** queries with **P** candidate pairs and S2AND has **X** clusters with **Y** author-paper pairs. **S**: Silver, **G**: Gold. SciDocs is evaluated per Cohan et al. (2020).

set of candidate papers by their relatedness to the query. Ad-hoc search is a critical mechanism for paper discovery in practice, and we gather multiple real-world data sets for training and evaluation. We include *TREC-CoVID* (Voorhees et al., 2021) and NFCorpus (Boteva et al., 2016) as test tasks for this format. We also introduce *Search*, a large training set of over 500K clickthrough events from Semantic Scholar, a scholarly search engine.

To evaluate the ad-hoc search tasks, candidate papers are ranked by increasing Euclidean distance between the query and the candidate embeddings. Pytrec_eval (Van Gysel and de Rijke, 2018) is used to calculate the ranking metrics. Normalized Discounted Cumulative Gain (nDCG) is used for Search and TREC-CoVID as the true relevance score can be $> 1$.

**Proximity** Similar to ad-hoc search, proximity tasks involve ranking a set of candidate papers by their relatedness to a query, except the query in this case is a paper as well. Proximity-based tasks form a basis for paper-based retrieval and recommendation, and for estimating paper similarity for use in applications like author disambiguation. We

include a total of ten proximity-based tasks, including four nearest neighbor tasks from SciDocs (predicting citations, co-citations, co-viewed or co-read papers), and three others from previous work: the *S2AND* author disambiguation task (Subramanian et al., 2021) with paper similarity features; *Paper-Reviewer Matching* (Mimno and McCallum, 2007; Liu et al., 2014; Zhao et al., 2022), where candidate reviewers are ranked by expert annotators based on the similarity of their papers to the query paper to be reviewed; and finally RELISH (Brown et al., 2019): a document recommendation task consisting of ground truth labels collected by a consortium of over 1500 scientists worldwide. We also introduce three new large-scale training datasets aimed at predicting same-authors, citations (via triplets) as in Cohan et al. (2020), and influential citations, which we define as four or more citations of the same paper in the text of a single paper.

For evaluation, we again rank candidate embeddings by Euclidean distance, using MAP and nDCG as the scoring metric except for S2AND with B$^3$ F1 (Bagga and Baldwin, 1998), and Paper-

Reviewer Matching, which uses precision@5 and @10.

**Classification** Classifying papers into topical categories is a foundational task for document organization and discovery. Apart from the two SciDocs tasks (*MAG* and *MeSH Diseases*), we have four others; including a binary task to predict whether a paper is relevant to biomimicry (Shyam et al., 2019), two biomedical classification tasks, namely *DRSM* from Burns (2022) and *MeSH Descriptors* classification (Lipscomb, 2000), and a new large-scale field of study (*FoS*) multi-label training set of more than 500K papers with silver FoS labels based on publication venues.

We evaluate embeddings on classification by scoring their performance as features within linear support vector classifiers. Results for these tasks are evaluated using binary/macro F1 score. To better understand how embeddings perform in data-scarce regimes, we also construct two few-shot versions each from Biomimicry, DRSM and FoS dataset subset for which we have manually annotated gold labels.

**Regression** We also consider regression tasks where the goal is to predict a continuous quantity for a given paper. For evaluation, we consider predicting three numeric attributes related to prominence or quality: *Tweet Mentions* (Jain and Singh, 2021), and two new datasets predicting peer review rating and maximum h-index of authors for a set of ICLR papers from OpenReview[1]. For training, we introduce two additional datasets; predicting citation count and publication year of papers.

We evaluate embeddings on regression by scoring their performance as features within linear support vector regression models. The reported results are computed as Kendall's $\tau$ rank correlation between the true and predicted labels.[2]

## 4 Multi-format representation learning

Typical approaches for learning document embeddings produce a single embedding for every task (Cohan et al., 2020; Ostendorff et al., 2022b). We hypothesize that a single embedding is insufficient for generalizing across multiple downstream tasks. At the other extreme, learning embeddings for each task separately limits generalization to new tasks

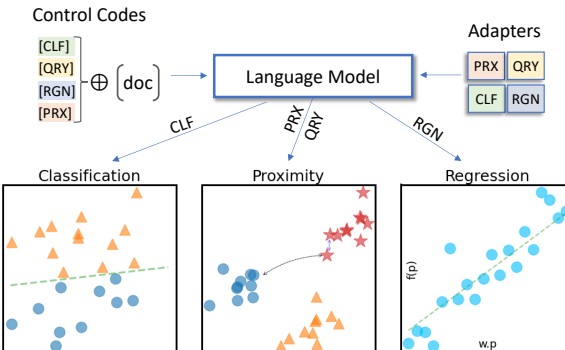

Figure 1: Generating multi-format embeddings. A task format is either associated with a control code appended to the input, or adapter blocks attached to the model.

and also incurs significant storage costs scaling with the number of tasks. We propose a method for learning a distinct document embedding for each task format, using a multi-task learning setup.

We assume we are given labeled data from a set of tasks for our four formats (ad-hoc search, proximity, classification, and regression), and we learn models capable of producing an embedding for any given (paper, format) pair. Our goal is for these embeddings to be used in lightweight classifiers/regressors as well as in nearest neighbor tasks, which we evaluate both on held-out data from the training tasks, *and* on new held-out tasks.

To help build intuition for why different embedding sets for different task formats may be helpful, Figure 1 illustrates the qualitative distinctions between the task formats. In general, an embedding space yielding good results for one task format may be less suited to others; for example, the classification space provides an error-free linear classifier, but its nearest neighbor pairs are not always of the same class. Empirically, we find that learning specialized embeddings per format improves performance, and that embeddings trained on a format tend to perform better on held-out tasks with the same format (see Table 3). Further, partitioning randomly (as discussed in section 7) was less effective than the format-based partitioning. Nonetheless, format-based partitioning is just one choice of many and experimenting with other partitioning schemes is an important item of future work.

### 4.1 Model

We follow Cohan et al. (2020) in using a pretrained transformer encoder as our base model. A scientific document is given as input to the encoder as a concatenation of its title and abstract separated by the

---

[1] https://api.openreview.net

[2] We found in our experiments that Pearson's $\rho$ and Kendall's $\tau$ produced similar relative results between models. We did not use MSE because its unbounded values could skew the overall average.

[SEP] token.[3] Unlike Cohan et al. (2020), we use three different types of training objectives suitable for each format to further train the model as described in subsection 4.2. We explore two methods to learn separate embeddings for each task form: control codes and adapters as shown in Figure 1.

**Control Codes** In this approach, we prepend a special token per-format ( Table 6 in the appendix) to the input and pass it to the transformer, taking the final layer embedding corresponding to this token as the document representation and feeding it to the task-specific head (described in subsection 4.2).

**Adapters** Adapters have been shown to be effective for multi-task learning, therefore we explore Adapter Fusion (Pfeiffer et al., 2021) and PALs (Stickland and Murray, 2019) in our work. Both these methods introduce task-specific adapters and attention modules at each transformer layer. Since we aim to learn different embeddings for different task formats, we create modules for each task format rather than each task, and the final embedding of the [CLS] token output via the adapter is taken as the corresponding representation of the document.

## 4.2 Training

We train our models in a multi-task setup with task-heterogeneous batching (Aghajanyan et al., 2021). For classification and regression, we use linear heads atop the base transformer.[4] Cross Entropy and Binary Cross Entropy loss with sigmoid activation are used for multi-class and multi-label classification respectively. For regression we minimize the Mean Squared Error loss.

For proximity and ad-hoc search tasks we use the triplet margin loss from Cohan et al. (2020). For these task forms, given a query, a relevance score accompanies each candidate. Hence, each training instance in this setup is a triplet consisting of a query $\mathcal{Q}$ in the form of a paper (for which we wish to find similar documents) or raw text, a positive candidate paper $\mathcal{P}+$ and a negative candidate $\mathcal{P}-$, where $\mathcal{P}+$ has a higher score than $\mathcal{P}-$. Then, we optimize the triplet loss:

$$L_{triplet} = \max\{d(\mathcal{Q}_E, \mathcal{P}_E^+) - d(\mathcal{Q}_E, \mathcal{P}_E^-) + \epsilon, 0\} \quad (1)$$

where $d$ is the Euclidean distance used as a measure of similarity between the query embedding $\mathcal{Q}_E$ and candidate embeddings $\mathcal{P}_E^+$ and $\mathcal{P}_E^-$. $\epsilon$ is the margin hyperparameter whose value is 1 chosen based on preliminary experiments.

## 5 Experiment Setup

**Training Data** Our multi-format models are trained on the 8 large in-train tasks detailed in Table 1. For proximity and ad-hoc search, we create up to 5 instances per query by sampling positive and negative papers from its candidate pool. We limit the number of training samples from each task to 600K,[5] resulting in training and validation sets of a of 3.27M and 446K instances respectively.

**Transformer Baselines** As a first step, we evaluate existing document representation methods on SciRepEval. These include SciBERT (Beltagy et al., 2019), a language model pre-trained on scientific corpora, and paper-embedding methods SPECTER (Cohan et al., 2020), ASPIRE (Mysore et al., 2022), and SciNCL (Ostendorff et al., 2022b) which is the state-of-the-art on SciDocs. ASPIRE produces representations for aspect-based matching between query and candidate papers which is a similar setting to our proximity tasks, so we only evaluate and report its results for that subset in Appendix C. Finally, we also evaluate two recent general-purpose text embedding methods, E5 v2 (Wang et al., 2022) and MPNet (Song et al., 2020). These methods are pre-trained on a set of over 1B query-candidate pairs including scientific text, and are two of the best BERT-base sized models on the recent MTEB benchmark (Muennighoff et al., 2022) for general-purpose text embeddings.

**SPECTER2 Models** Although SPECTER and SciNCL are strong baselines, about 70% of their pre-training data is from just two domains as shown in Table 9. This leads to poor generalization as outlined in Medić and Šnajder (2022) where BM25 outperformed both models across all the domains except Computer Science and BioMed on a citation recommendation task. To increase the domain coverage we pre-train a new model similar to Cohan et al. (2020), but with 10x more data spanning 23 fields of study and term it *SPECTER2 Base*. Each query paper in our training set can have up to 10 triplets where the positive candidates are taken from the direct citations of the query. 6 easy negatives are sampled based on the field of study (4 same as the query; 2 different) while 4 hard negatives come from papers cited by one of the query paper's citations but not by the query itself. We also include the SciNCL triplets as a subset. This

---

[3]In the Search task, publishing venue and year is also provided.

[4]The linear heads are discarded after training.

[5]Performance with smaller dataset samples - max 400K samples/tasks was relatively poor.

| Model | In-Train | Out-of-Train | Average |
|---|---|---|---|
| Transformer Baselines | | | |
| E5-base-v2 | 55.7 | 70.9 | 67.0 |
| MPNet | 49.0 | 71.0 | 65.3 |
| SciBERT | 51.5 | 60.2 | 58.0 |
| SPECTER | 54.7 | 72.0 | 67.5 |
| SciNCL | 55.6 | 73.4 | 68.8 |
| _SPECTER2_ | | | |
| Base | 56.3 | 73.6 | 69.1 |
| MTL CLS | 60.2 (0.44) | 72.1 (0.21) | 69.0 (0.19) |
| MTL CTRL | 62.4 (0.09) | 73.1 (0.18) | 70.4 (0.13) |
| Adapters | 62.4 (0.06) | 73.9 (0.13) | 70.9 (0.09) |
| PALs | 61.8 (0.27) | 72.6 (0.27) | 69.9 (0.2) |
| Fusion | 62.4 (0.08) | 73.9 (0.07) | 70.9 (0.04) |
| Adapters + MTL CTRL | **62.9** (0.09) | **74.1** (0.24) | **71.2** (0.19) |

Table 2: Evaluation results on SciRepEval in multiple settings. SPECTER2 Models are further categorized based on the training setup. Base is trained only on citations as mentioned earlier; MTL CLS generates a single embedding for all tasks, MTL CTRL (control codes) and Adapter variants (Adapters, PALs, and Adapter Fusion) produce an embedding per task format. We also consider an ensemble approach that averages the MTL CTRL and Adapter embeddings. For models we trained, we report the mean and standard deviation (in parentheses) across 5 runs with different seeds. The best results are highlighted in **bold**. We conduct one way analysis of variance (ANOVA) with Tukey's test (Haynes, 2013) for $\alpha = 0.05$ across multiple settings and underline those not statistically significantly different from the best.

results in 6.2M training and 176k validation triplets, and we release this data to the community.

Next, for our multi-format experiments, we pre-fine-tune the base model on SciRepEval in-train tasks both with (MTL CTRL) and without (MTL CLS) the control codes. Finally, to compare the control codes approach with adapters, we train the base model with BERT PALs and Fusion architectures. Fusion, being a two step process, first introduces task format specific adapters and then fusion modules. The MTL CTRL and adapter approaches produce multiple representations per document while MTL CLS produces a single representation. We use the PyTorch implementations of the models by HuggingFace[6]. The specific training configurations are described in Appendix B.

## 6 Results

Table 2 shows the evaluation of all our transformer baselines producing both single and multiple representations per document on SciRepEval. Our

[6]https://huggingface.co/models

benchmark includes diverse tasks with a variety of different evaluation metrics, and following previous work (e.g. Wang et al. (2019)) we report an average of the individual metrics (each ranging from 0-100). Among the vanilla models, even though E5 v2 and MPNet perform better than SciBERT; SPECTER, SciNCL and SPECTER2 Base outperform them suggesting the need for domain and task specific embeddings to do well on SciRepEval. The pre-fine-tuned multi-format variants of SPECTER2 outperform the baseline models on average. However, SPECTER2 Base and MTL CLS are on par even though the latter is trained on in-domain datasets. This is where the task format based training helps. We find that all the approaches that produce multiple representation types outperform the MTL CLS model, which learns only a single representation shared for all tasks by 1.4 to 2 points. The adapter variants are better than MTL CTRL overall, and result in an improvement of up to 0.8 points on the out-of-train tasks with the fusion adapters performing the best.

Further, as shown in Table 4, the control codes and adapters are the most efficient in terms of model size and computation runtime. Hence, we try to improve upon each by combining representations from both models by averaging them[7], and we find that these combined embeddings outperform the individual models consistently across the in-train, out-of-train, and overall average settings.

**Alternative Base Models** To test the consistency of our findings for the proposed training techniques, we also train the MTL CLS, MTL CTRL and adapters variants with SPECTER and SciNCL as the base models. Table 8 in Appendix D shows that the MTL CTRL token and the adapters approaches still substantially outperform the MTL CLS approach, suggesting that the efficacy of using an embedding per task format instead of a single embedding per document is consistent across a range of base model types.

## 7 Analyses

**Specialization of Control Code Embeddings** Our hypothesis is that by training embedding spaces on particular task formats, they will become more accurate for tasks of that format than for others. We test this by sampling one in-train and one

[7]We also tried concatenating both the embeddings, which yielded similar results with double the embedding size.

| Task format | Control Code Used | | | |
|---|---|---|---|---|
| | CLF | RGN | PRX | QRY |
| Classification | 43.3 | 29.4 | 32.7 | 31.1 |
| Regression | 29.8 | 46.8 | 43.3 | 43.1 |
| Proximity | 87.4 | 78.9 | 88.8 | 87.5 |
| Search | 73.4 | 72.6 | 76.1 | 78.5 |

(a) in-train

| Task format | Control Code Used | | | |
|---|---|---|---|---|
| | CLF | RGN | PRX | QRY |
| Classification | 64.8 | 63.6 | 62.8 | 63.7 |
| Regression | 16.9 | 22.2 | 17.8 | 16.1 |
| Proximity | 43.8 | 40.5 | 45.1 | 45.2 |
| Ad-hoc search | 87.4 | 83.1 | 90.3 | 90.9 |

(b) out-of-train

Table 3: Cross task analysis for control codes. The best results for each task format across all control codes is underlined. These are represented in the diagonal for both in-train and out-of-train tasks suggesting that format based partitions in multi-task training produce representations suitable for the corresponding format.

out-of-train[8] task of every format (for ease of computation) and evaluating them with *all* the control codes. As shown in Table 3, the control codes trained on a task format yield best results for tasks of that format, for both in-train and out-of-train.

As an extension to this we also analyze how well the control code representations work when the training tasks are randomly grouped together as opposed to by task format. From an evaluation across 5 random partitions, we found that task-format-based partitions were better by over 2.7 points on average (both in-train and out-of-train) across the formats, suggesting that representations specific to each task format do lead to better results overall.

Finally, to study training affinity among the task formats themselves, we pre-fine-tune on at most two formats at once. Appendix H reveals that combined multi-task training on similar task formats like proximity/adhoc-search results in performance gains, but only on the related tasks. Training on all the tasks yields better results on average across the task formats.

**Efficiency** While the variants producing task-format based representations are strong baselines on SciRepEval as shown in Table 2, efficiency is an important practical consideration. As shown in Table 4, the control code approach only requires one new control code embedding per format, and

[8]In-train: FoS, Citation Count, Same Author Detection, Search; Out-of-train: DRSM, Peer Review Score, Peer-Reviewer Matching, TREC-CoVID

| Model | Parameters per Task Form | Training Time | Inference Time |
|---|---|---|---|
| MTL CTRL | 768 | 1x | 1x |
| PALs | 2M | 1.42x | 1.29x |
| Adapters | 1M | 0.96x | 1.05x |
| Fusion | 22M | 1.32x | 1.69x |
| Adapters + MTL CTRL | 1M | 1.96x | 2.05x |

Table 4: Parameter and (relative) runtime efficiency of SPECTER2 models. MTL CTRL and Adapters are similar in runtime, but PALs, Fusion and ensemble variants add significant computation costs.

| Model | MAP | R@5 |
|---|---|---|
| BM-25 | 33.7 | 28.5 |
| SPECTER2 Base | 38.0 | 32.4 |
| SPECTER2 MTL CLS | 34.6 | 24.9 |
| SPECTER2 MTL CTRL | 36.5 | 30.7 |
| SPECTER2 Adapters | **38.4** | **33.0** |
| SPECTER2 Adapters + MTL CTRL | **38.4** | 32.9 |

Table 5: Comparison of SPECTER2 models with BM25 on the MDCR benchmark. As in the original paper, we report MAP and Recall@5 scores. The best results obtained are highlighted in **bold**.

has no impact on training time. PALs, in contrast, introduce new attention layers and train the entire network, increasing runtime, while Adapters add and only train half as many parameters as PALs. Fusion layers have 10x as many parameters as PALs leading to 2x more time on inference. The ensemble model Adapters + MTL has the highest training and inference cost, so the cheaper Adapters, achieving only slightly lower task performance (Table 2), may be preferable in many use cases. Training and inference times are measured with 1k and 10k samples, respectively. As such, we release the SPECTER2 Base model and our best task-format Adapters publicly for further use.[9]

**MDCR** Medić and Šnajder (2022) introduced the new MDCR citation-recommendation benchmark showing that BM25 outperforms neural models on the task of retrieving papers cited by a query paper from a large candidate pool across multiple scientific fields. Table 5 shows that our multi-task format based training establishes a new state of the art on the benchmark, with the Adapters yielding the best results. We report results broken out by different fields of study in Appendix E.

[9]https://github.com/allenai/SPECTER2

# 8 Conclusion

We introduce SciRepEval, a benchmark for scientific document representation methods with 24 tasks across four task formats. On this benchmark, we show that learning a separate document representation for each task format substantially improves task performance compared to learning a single representation for all tasks. Future work could address limitations of our work by evaluating partitioning schemes beyond task format, crafting higher-fidelity metrics to account for the diversity of tasks in SciRepEval (which vary in sensitivity and in relevance to downstream applications), or further exploring how accuracy varies with computational and storage cost.

## Limitations

**Dependence on short text features** All the tasks and training techniques described in our work depend upon the textual features of a scientific paper, namely titles and abstracts. Depending upon the document source the abstract may not always be present. To be robust to this case, when pre-training SPECTER2 Base we ensure that for 5% of the papers we only process the titles. However, the model performance may be subpar in the absence of the paper abstract.

Further, we do not explore methods that use full text of papers since this is less consistently available (and also increases computational cost of our transformer-based models), but exploring this could be helpful in future work, Moreover, scientific documents are associated with rich metadata other than the text contents such as authors, citations, venue, etc. We use this data in some tasks like *Search* and *Citation Prediction*, but not in others. The data can be useful in the absence of paper text, and as part of future work we can explore supplementing the textual features.

**Potential benchmark extensions** A number of improvements to our benchmark are possible, including:

- **Fields of study:** The large training set contains silver labels automatically derived from publication venues with only a few hundred manually annotated samples for evaluation. The gold label proportion could be increased, and/or improved silver data can be obtained from, for example, high-quality large language models.

- **S2AND:** The evaluation for this task requires setting up a separate project repository (Subramanian et al., 2021). Our current implementation only produces the input embeddings for this task, but future work could integrate the entire evaluation workflow within the benchmark.

- **Peer Review Score, h-Index of Authors:** The data for these tasks is sourced from OpenReview and consists of a static set of papers from 2017-2022. The data could be periodically updated to avoid staleness, and to analyze yearly trends.

- **Cite (SciDocs):** About 1700 positively labeled citation pairs in the task (Cohan et al., 2020) are in fact citations of citations. Since it is an existing benchmark, we include it without any changes, but we could extend SciRepEval by including MDCR (Medić and Šnajder, 2022) which covers more fields of study, but we do report MDCR results in this work.

- **More task formats:** Although proximity embeddings from the SPECTER2 models can be used for any task formats not covered by SciRepEval, we could extend the benchmark to include more formats such as question answering and generation.

- **Confirming benchmark findings in real-world applications:** Our experiments show that the new embeddings we introduce raise performance on our benchmark; it would be helpful to determine whether the higher scores on our fixed data sets actually translate to better application metrics, by performing e.g. an online A/B test with an application that consumes the embeddings (paper recommendation, search, etc.).

**Other task partitioning methods** We partition the SciRepEval training tasks as per task formats based on the intuition in section 4 and although it gives better performance than 5 random partition runs as reported in section 7, there may be other task partitions that can yield even better results. Clearly, including the task format to be evaluated on during training helps as seen in Table 11, but this still does not return the best overall results on our benchmark. During our preliminary experiments, we had an additional format specialized to author-based tasks which included *Same Author*

*Detection*, but found that incorporating this task into the proximity tasks instead led to better performance. Evaluating other task partitions is an item of future work.

## Acknowledgements

We would like to thank Jonathan Bragg, Zhipeng Hou, and the anonymous reviewers for helpful comments, suggestions and feedback. We would also like to acknowledge the support of NASA AATT Project for funding the PeTaL research and contributing the biomimicry dataset. This work was supported in part by NSF Grant 2033558.

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

# A   SciRepEval Tasks

## A.1   Ad-hoc search

**Search**   We used click-through data from Semantic Scholar, an academic search engine. Only search queries with at least 10 results were included, and a set of heuristic rules were applied to exclude likely noise and bots. We removed author queries when the query was judged to contain any person tokens by named entity recognition (Honnibal et al., 2020).

**NFCorpus**   This is a set of non technical English queries and associated results compiled from NutritionFacts.org (Boteva et al., 2016). The candidate document are selected from PubMed and the relevance score represents whether the candidates are linked from the query article webpage. A score of 2 represents a direct link and 1 represents a link from a directly linked article. We use the test subset of 323 queries for evaluation. To generate the negative candidates, we randomly sample at most 100 documents for each query from the remaining Pubmed corpus.

**TREC-COVID**   TREC-COVID was introduced by Voorhees et al. (2021) as a biomedical literature search task relevant to COVID-19. The dataset consists of 50 search queries and candidate literature from the CORD-19 corpus (Wang et al., 2020b) along with their relevance scores on a scale of 0-2.

Each query consists of a short title, a question asking the required information and a narrative describes briefly exactly the type of information that the results should have. For our evaluation we combine these fields into a single text separated by the [SEP] token.

## A.2   Proximity

**S2AND and Same Author Detection**   The S2AND dataset (Subramanian et al., 2021) contains signatures (author-paper pairs) that are clustered according to which author mentions refer to the same person. Due to the high resource requirements of running the original S2AND evaluation, we create S2AND-mini, a version of S2AND with only 1000 blocks from each of S2AND's dataset sources and at most 500 signatures per block. Our evaluation of S2AND-mini follows the original evaluation of S2AND; that is, our method's document embeddings are used along with author and paper metadata to create features for a clustering algorithm that consists of a pairwise scoring model followed by greedy agglomerative clustering. We use $B^3$ F1 (Bagga and Baldwin, 1998) as in the original paper for evaluation.

We also use S2AND to create the data for our same-author detection task. Unlike the original S2AND evaluation, our same-author task uses only paper embeddings without any additional author or paper metadata, which allows us to directly train the embedding model on the data. Same-author detection is formulated as a triplet ranking task; given three papers of which two share an author, the goal is to find the matching pair.

**RELISH**   An acronym for RElevant LIterature SearcH, this task is a joint annotation effort of more than 1500 scientist across 84 countries (Brown et al., 2019). It consists of over 190k relevance labels for PubMed articles with regards to a given query (seed) article. The articles cover 76% of MeSH descriptors. The labelling scheme has 3 scores: 2-relevant, 1-partially relevant and 0-irrelevant. We use the complete dataset for evaluating our methods.

**Peer Reviewer Matching**   In this task the goal is to judge whether a given paper is relevant to a potential reviewer. As data for this task is hard to obtain at scale due to the double-blind nature of many conferences and journals, we combine multiple existing reviewer-paper matching datasets:

- Mimno and McCallum (2007), with 393 paper-review relevance ratings from a corpus of 148 NeurIPS 2006 papers and 364 reviewers, annotated by nine human experts.

- Liu et al. (2014), an extension of Mimno and McCallum (2007) which adds 766 additional paper-review annotations.

- Zhao et al. (2022), with 694 paper-reviewer relevance ratings from a corpus of 75 papers and 1833 reviewers from the IEEE ICIP 2016 conference, annotated by 3 human experts.

All datasets have been annotated on the same 0-3 relevance rating scale. The candidate reviewers are all researchers, and we embed all the papers written by them using our models. To obtain the model's score for each candidate reviewer, we compute the cosine similarity between the query paper and each of the candidate's papers, and take the mean of the top 3 similarities as the score. We consider two ways to map the 0-3 relevance judgements to binary labels—hard and soft decision—where for the soft decision a score of 2 or 3 is considered relevant and for hard decision only a score of 3 is considered relevant. Precision at 5 (P@5) and 10 (P@10) results are used as the final metric, which ultimately results in four numbers (P@5 and P@10 for each of hard and soft decisions), which are averaged to produce the single number reported in our final results for this task.

**Highly Influential Citations**    In this task, given a paper $A$ and paper $B$, we aim to predict whether $B$ is highly influenced by $A$. As measuring influence is subjective and human annotation is expensive, we approximate influence by counting the number of times $A$ is cited in the text of $B$. If $A$ is cited at least 4 times, we consider it to be highly influential (a positive example in our triplet-based loss); otherwise, we consider it to be a negative example. During evaluation, we sample query papers which have at least 5 positive candidates and compute the L2 distance for similarity ranking. Note that our definition of 'influential' differs from that in Valenzuela et al. (2015).

**Citation Prediction (SPECTER Pre-training Triplets)**    This is the task and dataset used for pre-training in Cohan et al. (2020). It is based on citation links between scientific documents where each instance is a triplet consisting of a query, a positive and a negative paper. Each query can have

up to five triplets, where the positives are sampled from papers directly cited by the query and negatives are chosen either randomly (easy) or from citations of citations (hard). 3 easy and 2 hard difficult are chosen for each query. To evaluate the effectiveness of this pre-training we follow Cohan et al. (2020) and use SciDocs for evaluation, excluding the recommendations task.

## A.3   Classification

**MeSH Descriptors**    Medical Subject Headings (MeSH) (Lipscomb, 2000) indexes biomedical publications into a categorical hierarchy consisting of descriptors which refer to topic headings and specific aspect related to a topic respectively. The dataset is a collection of scientific documents belonging to the 30 most frequently occurring top level MeSH descriptors and having exactly one qualifier. We filter out the records that don't have an associated qualifier. The descriptors thus serve as the labels in the multi-class classification task.

**Fields of Study (FoS)**    The FoS task is a multi-label classification problem where each scientific document is assigned one or more classes out of 23 possible fields. For gold test data, we manually labeled 471 papers into at most three fields-of-study. For silver training data, we assumed that a paper within a venue generally falls within a narrow set of fields and manually assigned FoS labels to publication venues. We then propagated the venue labels to the papers published therein.

To evaluate different data sizes, we obtain the F1 score on the gold data in three settings: 5-shot, 10-shot, and the complete gold test set. The average of these scores is treated as the score for this task when computing the overall average score for the benchmark.

**Disease Research State Model (DRSM)**    DRSM (Burns, 2022) is a collection of Pubmed papers that deal with six specific aspects of rare diseases. The gold data is annotated by in-house experts and used for evaluation, while the silver data is generated by annotation service providers with medical expertise.

Similar to FoS, we obtain the F1 score on 24-shot, 64-shot, and full data, then average the results before computing the final benchmark score.

**Biomimicry**    We sample tags for a set of papers in the PeTaL database (Shyam et al., 2019) to create a binary classification dataset with labels indicating whether each paper is about biomimicry. The data

is unbalanced, with only 13% positive samples. We evaluate 16-shot, 64-shot, and full-data setup and take the mean to get the final score.

## A.4 Regression

**Citation Count**   We sample a collection of scientific articles published in 2016 from the set of papers in the search dataset described in section A.1, so that a 5 year period has passed for them to collect citations. Each article has at least one citation, and the citation counts are converted to log scale.

**Year of Publication**   The aim of this task is to determine research trends by predicting the year of publication of a scientific article. We sample publications from the search dataset with a publication date after the year 2005 and scale the years so that their values are between 0 and 1. Further, since this task is used for training along with citation count prediction, and to align the loss scales, the labels are scaled by the mean of the labels in citation count for parity.

**Peer Review Score**   We use the OpenReview API[10] to collect paper metadata and corresponding review scores for ICLR conferences from 2017 to 2022. Each reviewer in ICLR assigns a final rating in the range [0-10], and we take the mean rating as the label for every paper.

**h-Index of Authors**   In this task the goal is to predict the maximum h-Index of any of the authors of a scientific publication. We re-use the peer review score dataset, obtain the h-Index of all the authors for each paper using the Semantic Scholar API[11], and pick the max as the label. The labels are normalized to lie between [0,1].

**Tweet Mentions**   The goal of this task is to predict the combined number of a paper's mentions and retweets. We post-process the dataset created by Jain and Singh (2021) containing tweets about Arxiv papers between 2010-19. The sum of normalized counts of mentions and retweets is finally considered as the score to be predicted.

The prior versions of SciRepEval included three feeds datasets. We observed high correlation in model performance among these tasks, similarly to SciDocs in Appendix G. Moreover, each query was associated with 10 candidates on average, which is small for a real recommendation engine. For a more robust evaluation with larger candidate pools,

| Task form | Input format |
|---|---|
| Classification | concat([CLF],doc) |
| Regression | concat([RGN],doc) |
| Proximity | concat([PRX],doc) |
| Ad-hoc Search | concat([QRY]/[PRX],query/doc) |

Table 6: Assigned input formats and control codes for each task form. [CLF], [RGN], [PRX] and [QRY] are special tokens, doc is the input.

the feeds datasets have been replaced with REL-ISH and NFCorpus. Both are recommendation tasks similar to feeds – RELISH, a proximity task with 60 candidates/query and NFCorpus, an adhoc-search task with 138 candidates/query on average.

## B   Implementation details

During pre-training, all the tasks with the same format share their task-format specific parameters. The control code based paradigm introduces four new (randomly-initialized) special tokens to the vocabulary. We try initializing these additional parameters randomly, with the [CLS] token and a combination of [CLS] with some noise. However, it has little impact on the resulting model performance with random initialization being better on average. Further, we also tried loss weighting strategies (Chen et al., 2018; Liu et al., 2019a) but our preliminary experiments produced better results without any scaling so we didn't explore it further. All the base models are trained for two epochs on two 48GB NVIDIA Quadro RTX 8000 GPUs with 16 bit precision, an effective batch size of 256, and a maximum input length of 512 tokens. Each batch is sampled with an equal number of examples from each task.[12] We use AdamW (Loshchilov and Hutter, 2019) with $\epsilon = 1e\text{-}8$. The learning rate follows an inverse square root schedule with a linear warmup of 700 steps and peak of 5e-5.

The adapter approaches follow the two step training process and learning rate configurations described in Pfeiffer et al. (2021). One adapter per task family is attached to the base model in both single adapter and fusion stages and is trained for a maximum of 6 and 4 epochs respectively. For PALs one layer is added per task format and the entire network is trained for 2 epochs as in Stickland and Murray (2019).

---

[10]https://api.openreview.net
[11]https://api.semanticscholar.org/

[12]We experimented with mixed and task sequential batching as well which did not yield good results.

| Model | In-Train | Out-of-Train | Avg |
|---|---|---|---|
| TS ASPIRE$_{CS}$ | 65.0 | 86.3 | 82.7 |
| TS ASPIRE$_{Bio}$ | 65.5 | 86.0 | 82.6 |
| OT ASPIRE$_{CS}$ | 64.5 | 86.4 | 82.8 |
| OT ASPIRE$_{Bio}$ | 65.0 | 86.0 | 82.5 |
| *SPECTER2* | | | |
| MTL CTRL | 67.0 | 86.6 | 83.3 |
| Adapters | 67.7 | 86.8 | 83.6 |

Table 7: Comparison of SPECTER2 multi-format methods with ASPIRE on proximity tasks. The best results for each base model are underlined. TS: Text Supervision, OT: Optimal Transport

## B.1 Evaluation

For classification and regression, we train a linear SVM on each downstream task using the embeddings as input, and we tune the regularization parameter $\mathcal{C}$ via grid search. Multi-class and multi-label classification are configured under the one vs all classifier setting.

## C ASPIRE Evaluation

ASPIRE (Mysore et al., 2022) produces representations for the dense retrieval of scientific documents based on matching multiple aspects between the query and candidates. To evaluate these representations under the settings they are designed for, we only report the results on the proximity tasks in Table 7. We use the model implementations available on HuggingFace which have been pre-trained on documents from the Computer Science (CS) and Biomedical (Bio) domains. The models variants can be further sub-categorized as retrieval based on best aspect matching (TS ASPIRE) and a weighted sum of the similarity score among all the aspects based on Optimal Transport (OT ASPIRE) between the query and candidates. Both our multi-format approaches with control codes and adapters produce better results overall and on out-of-train tasks. Note however, since ASPIRE models are trained on co-citations, they perform much better on average on the citation based tasks from SciDocs and thus achieve relatively higher scores on out-of-train.

## D Robustness of multi-format training

Table 8 shows a comparison of SciRepEval results between MTL CLS, MTL CTRL and Adapter variants trained with SciNCL and SPECTER as the base models. The multiple embedding approaches of control codes and adapters are better than simple multi task learning that produces a single document embedding across the board.

| Model | In-Train | Out-of-Train | Average |
|---|---|---|---|
| *SPECTER* | | | |
| MTL CLS | 60.0 | 71.6 | 68.6 |
| MTL CTRL | 61.9 | 72.7 | 69.9 |
| Adapters | 61.5 | 73.2 | 70.2 |
| Adapters + MTL CTRL | 62.2 | 73.6 | 70.6 |
| *SciNCL* | | | |
| MTL CLS | 60.1 | 71.9 | 68.8 |
| MTL CTRL | 62.1 | 72.9 | 70.1 |
| Adapters | 61.9 | 73.8 | 70.7 |
| Adapters + MTL CTRL | 62.5 | 74.0 | 71.0 |

Table 8: Results for multi-format training with SPECTER and SciNCL as base models. For brevity, we report only the single adapters results due to their advantage of computation efficiency. The best results for each base model are underlined.

## E SciRepEval Domain Distribution and MDCR Evaluation

We study the domain diversity of SciRepEval and display the results in Table 9. To compare against the training data for SciDocs, we consider the citation prediction triplets on which SPECTER is trained which is also a subset of the SciRepEval in-train tasks. Even though Medicine and Computer Science papers still form a bulk of the data, SciRepEval has 105x more documents on average per domain compared to the SPECTER triplets.

Further, as shown in Table 5, our task format based models outperform BM25 on the MDCR benchmark (Medić and Šnajder, 2022) and establish the new state of the art. Table 10 displays the breakdown of the results by fields of study. Apart from Geology and History, the ensemble model is equivalent or better than BM25 on all the scientific domains.

## F SPECTER Objective

Lastly, we perform an ablation study to better understand the importance of the unsupervised citation-based training objective. We used SciBERT as the base model for this ablation since both SPECTER and SciNCL were trained with the citation objective. Removing the citation objective and its accompanying data from SciBERT + MTL CTRL, we find that the in-train performance drops from 61.9 to 61.8, while out-of-train drops from 57.9 to 57.5, hinting that the citation objective may be helpful for generalization to new tasks.

| Field of study | SciRepEval (A) | SciDocs (B) | Increase Ratio (A/B) |
|---|---|---|---|
| Medicine | 3,201,323 | 74,685 | 43 |
| Computer Science | 1,187,689 | 199,664 | 6 |
| Biology | 882,357 | 13,377 | 66 |
| Chemistry | 508,056 | 3,813 | 133 |
| Psychology | 492,071 | 22,590 | 22 |
| Materials Science | 271,865 | 7,681 | 35 |
| Engineering | 254,826 | 31,444 | 8 |
| Mathematics | 231,482 | 25,800 | 9 |
| Physics | 217,670 | 7,285 | 30 |
| Business | 217,585 | 5,450 | 40 |
| Sociology | 156,128 | 2,305 | 68 |
| Political Science | 154,388 | 1,032 | 150 |
| Economics | 123,357 | 2,705 | 46 |
| Environmental Science | 91,682 | 1,136 | 81 |
| Art | 89,527 | 206 | 435 |
| Geography | 83,688 | 1,491 | 56 |
| Philosophy | 61,996 | 151 | 411 |
| Geology | 51,103 | 640 | 80 |
| History | 46,430 | 159 | 292 |

Table 9: Data domain distribution in SciRepEval for the training tasks and comparison with SciDocs. We group the unique documents in both the benchmarks by their MAG (Wang et al., 2020a) fields of study and present the counts in columns 2 and 3 and the absolute increase per field in column4.

| | BM25 | | SPECTER2 Base | | SPECTER2 MTL CTRL | | SPECTER2 Adapters | | SPECTER2 Adapters + MTL CTRL | |
|---|---|---|---|---|---|---|---|---|---|---|
| **FoS** | MAP | R@5 | MAP | R@5 | MAP | R@5 | MAP | R@5 | MAP | R@5 |
| Art | 38.2 | 32.3 | 43.4 | **37.8** | 39.9 | 32.6 | **44.7** | 37.4 | 43.5 | 36.7 |
| Biology | 38.3 | 33.6 | 39.9 | 33.3 | 40.6 | 33.9 | 41.6 | 36.0 | **42.2** | **37.1** |
| Business | 28.1 | 22.5 | 35.0 | **30.5** | 32.7 | 28.6 | **35.3** | 29.6 | 34.7 | 29.3 |
| Chemistry | 38.0 | 32.6 | 39.7 | 33.5 | 38.9 | 32.4 | **41.4** | **35.7** | 40.8 | 35.3 |
| Computer Science | 34.8 | 30.5 | 38.5 | 33.4 | 38.9 | 32.8 | 39.3 | 33.5 | **39.9** | **34.7** |
| Economics | 30.5 | 26.0 | 33.7 | 28.5 | 31.9 | 26.3 | **33.7** | 27.3 | 33.7 | **29.5** |
| Engineering | 34.6 | 29.3 | 35.4 | **30.3** | 36.5 | **30.3** | 35.6 | 29.8 | **36.6** | 30.1 |
| Environmental Science | 31.6 | 26.2 | 35.0 | 27.9 | 36.8 | 30.7 | 36.7 | 30.5 | **38.1** | **32.5** |
| Geography | 31.8 | 27.8 | 37.1 | 31.8 | 34.0 | 29.2 | **37.2** | **33.3** | 36.7 | 32.6 |
| Geology | 33.1 | 28.0 | 33.4 | 27.7 | 32.5 | 25.8 | 33.6 | **28.4** | **34.1** | **28.4** |
| History | 38.1 | 32.9 | **41.9** | 34.7 | 37.3 | 31.3 | 41.4 | **36.1** | 40.3 | 34.7 |
| Materials Science | 36.1 | 30.7 | 39.7 | 34.0 | 39.4 | 32.8 | 39.9 | **34.3** | **40.7** | **34.2** |
| Mathematics | 35.3 | 28.3 | 40.8 | 34.2 | 39.0 | 33.6 | **41.7** | **35.7** | 40.8 | 34.2 |
| Medicine | 38.6 | 32.5 | 43.8 | 39.0 | 43.5 | 38.2 | **45.5** | **40.0** | 45.3 | 39.6 |
| Philosophy | 30.2 | 25.7 | **37.2** | 32.0 | 33.3 | 28.6 | 36.3 | **32.4** | 36.2 | 31.6 |
| Physics | 35.1 | 30.2 | 37.6 | 32.5 | 37.2 | 30.8 | 37.6 | 32.7 | **39.0** | **33.9** |
| Political Science | 28.6 | 23.1 | **35.7** | **31.6** | 32.0 | 26.9 | 35.0 | 29.5 | 34.0 | 29.0 |
| Psychology | 32.5 | 28.9 | 38.8 | 33.2 | 37.1 | 31.6 | **39.5** | **34.4** | 39.4 | 34.1 |
| Sociology | 26.8 | 20.5 | **34.6** | **29.8** | 31.2 | 26.2 | 34.3 | 29.9 | 33.5 | 28.0 |
| Avg | 33.7 | 28.5 | 38.0 | 32.4 | 36.5 | 30.7 | **38.4** | **33.0** | 38.4 | 32.9 |

Table 10: MDCR benchmark results breakdown by Fields of study (FoS). Best results are highlighted in **bold**.

## G Cross-Task Correlation Analysis

Figure 2 show's Pearson correlations of model performance metrics between tasks in SciRepEval. To compute the correlations, we include all of the individual task results of the model runs shown in Table 2 and Table 8, excluding the ensembles. The correlations between tasks in SciDocs (bottom right) are highest, while correlations between tasks in the entirety of SciRepEval span a larger range. Notably, DRSM, Biomimicry and S2AND are uncorrelated with most other tasks. In short, between-task diversity is larger in SciRepEval than in SciDocs.

## H Related Tasks for MTL Training

When training on multiple tasks simultaneously, it is important to choose a combination of tasks that does not display negative transfer (Aribandi et al., 2022; Padmakumar et al., 2022; Fifty et al., 2021). Given $\mathcal{T}$ tasks, it may be computationally prohibitive to train all $2^{\mathcal{T}} - 1$ task combinations to find the best one. Alternatively, recent work suggests pre-fine-tuning on a large collection of tasks simultaneously offsets the negative transfer between a subset of those tasks (Aghajanyan et al., 2021; Aribandi et al., 2022). Padmakumar et al. (2022) show that pre-fine-tuning on a small set of tasks related to the downstream task is more efficient than large scale multi-task training and yields similar results. As shown in Table 11, we study the training affinity among our task formats by pre-fine-tuning on individual task formats as well pairwise combinations in a multi-task setup. Supporting the findings in Padmakumar et al. (2022), pre-fine-tuning on the tasks from each individual format's training data gives better or similar performance on downstream tasks when compared to training on all the formats at once, except on out-of-train classification and search. Surprisingly, regression when combined with proximity and search gives the best result on both in-train and out-of-train tasks. Additionally, related tasks like proximity and ad-hoc search also provide one another a boost when trained together. However, training on all the tasks simultaneously yields the best results *on average*, and individual results are within 1 point of the best combinations per task format, except for out-of-train regression and in-train classification.

| Task Format(s) Trained On | Task Format Evaluated On | | | | | | | | | | |
|---|---|---|---|---|---|---|---|---|---|---|---|
| | *In-Train* | | | | | *Out-of-Train* | | | | | *All* |
| | CLF | RGN | PRX | SRCH | Avg | CLF | RGN | PRX | SRCH | Avg | Avg |
| CLF | **69.1** | 16.3 | 58.4 | 71.3 | 54.9 | 70.7 | 18.3 | 79.7 | 73.4 | 68.0 | 63.7 |
| RGN | 48.8 | **45.8** | 53.1 | 69.6 | 52.7 | 63.9 | **20.4** | 61.5 | 67.4 | 56.4 | 55.3 |
| PRX | 57.5 | 33.8 | 66.8 | 73.1 | 53.3 | 69.5 | 17.6 | **87.3** | 78.5 | 72.4 | 68.0 |
| SRCH | 53.2 | 32.0 | 59.8 | 78.1 | 53.4 | 63.5 | 14.6 | 77.6 | 78.0 | 65.4 | 62.0 |
| CLF+RGN | 68.4 | 41.9 | 60.9 | 71.8 | 54.6 | 71.5 | 18.0 | 79.0 | 73.9 | 67.8 | 65.6 |
| CLF+PRX | 65.9 | 31.9 | 65.7 | 73.0 | 56.5 | 70.9 | 16.3 | 86.6 | 75.7 | 71.8 | 68.0 |
| CLF+SRCH | 66.6 | 20.3 | 60.8 | 78.4 | 53.7 | 70.9 | 13.9 | 81.5 | 72.0 | 68.3 | 64.4 |
| RGN+PRX | 58.4 | 44.0 | **67.4** | 72.4 | 57.4 | 69.5 | 18.7 | **87.3** | 78.2 | 72.6 | 69.0 |
| RGN+SRCH | 54.6 | 44.4 | 60.2 | **78.8** | 55.6 | 67.2 | 18.4 | 77.2 | 78.3 | 66.5 | 63.9 |
| PRX+SRCH | 57.5 | 33.2 | 66.8 | 78.1 | 56.5 | 68.2 | 18.4 | 86.8 | 79.3 | 72.1 | 67.9 |
| All | 67.5 | 44.8 | 67.0 | 78.3 | **62.4** | 71.7 | 19.3 | 87.2 | 79.7 | 73.1 | **70.4** |

Table 11: Task relatedness analysis for choosing a sub-group of tasks to train on so as to obtain optimum performance. SPECTER2 Base model is trained on one or more task formats (rows) and then evaluated for a comparison with MTL CTRL (last row). Both per task format and overall average performance is reported (columns). The best training combination for every task is highlighted in **bold**. The best single and combined training results for every evaluated task format respectively are underlined.

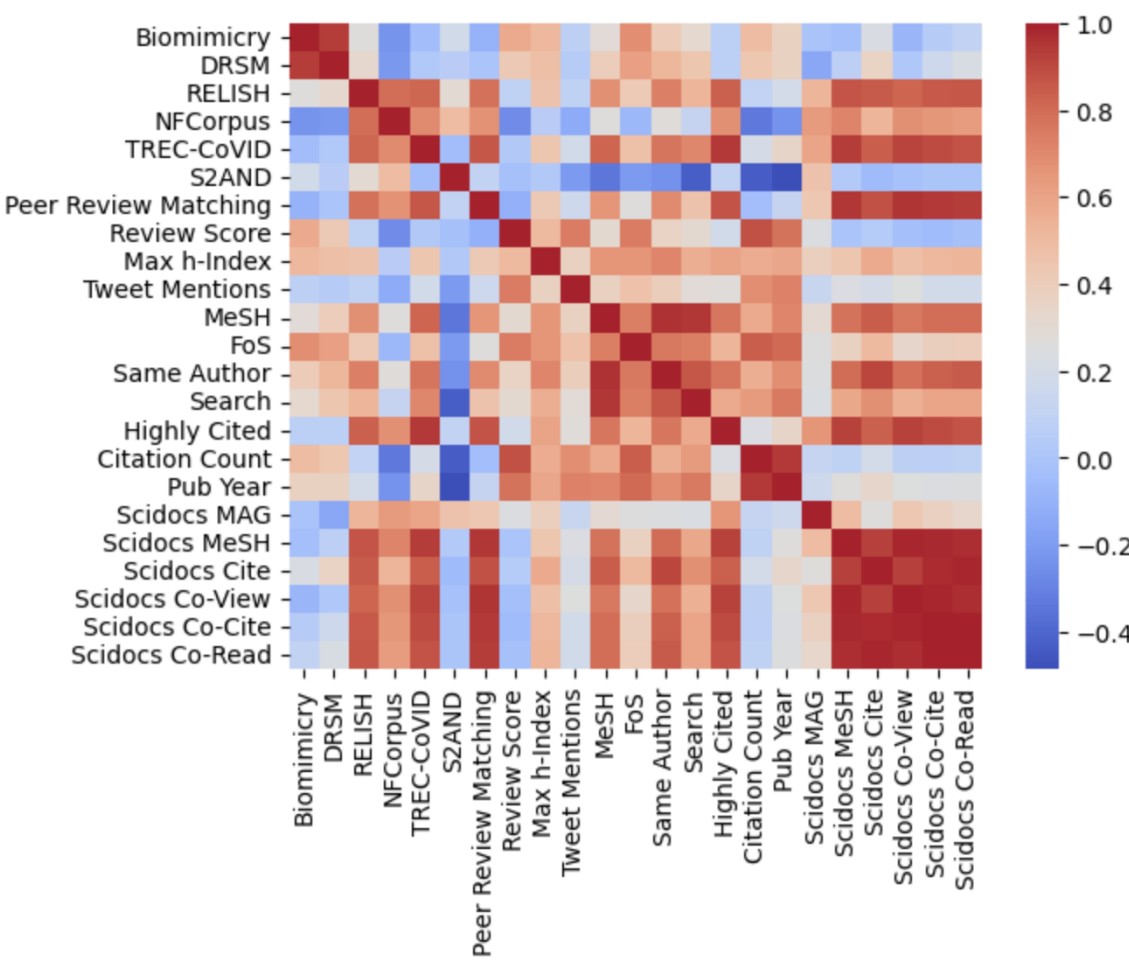

Figure 2: Correlations of model performances between tasks in SciRepEval.