# OpenReview forum: "SciRepEval: A Multi-Format Benchmark for Scientific Document Representations"
_EMNLP/2023/Conference — EMNLP 2023 Main_

### Official Review · Reviewer_U5Yz · 2023-08-05

**Soundness:** 4

**Excitement:**

3: Ambivalent: It has merits (e.g., it reports state-of-the-art results, the idea is nice), but there are key weaknesses (e.g., it describes incremental work), and it can significantly benefit from another round of revision. However, I won't object to accepting it if my co-reviewers champion it.

**Paper Topic And Main Contributions:**

This paper focuses on learning representations of scientific documents and introduces the first comprehensive benchmark for training and evaluating document representations. The authors use this benchmark to evaluate existing methods, demonstrating the tasks' challenges.

I'm not well-versed in scientific documents representation literature, so my comments should be taken with a grain of salt.

**Questions For The Authors:**

- Maybe I missed something, but I find the motivation of multi-format representing learning lacking. Could the authors elaborate on why this is important? Thanks!
- While there are benefits to learning separate representations for each task format, the cost of storing separate models could be substantial. The authors elaborated on the inference cost in Section 7, but I'm wondering if there are other impacts on, for example, storage.


**Reasons To Accept:**

- A new and solid benchmark that could be useful for the research community.
- Well-executed experiments.
- The authors also introduce methods that improve existing methods on the proposed benchmarks.


**Reasons To Reject:**

- This is not necessarily a reason for rejection, but this paper is packed with contents (dataset, evaluation, and new methods). The authors could consider splitting the content of the paper into separate papers.


**Reproducibility:**

4: Could mostly reproduce the results, but there may be some variation because of sample variance or minor variations in their interpretation of the protocol or method.

**Reviewer Confidence:**

3: Pretty sure, but there's a chance I missed something. Although I have a good feel for this area in general, I did not carefully check the paper's details, e.g., the math, experimental design, or novelty.

---

> ### Author Rebuttal · Authors · 2023-08-29
>
> We appreciate the helpful comments from the reviewer.
>
> Answers from Authors:
>
> 1. The motivation for multi-format representation learning stems from the need for a single scientific document representation model that can generalize well across a wide range of tasks formats without further expensive fine-tuning. As also noted in response to review 2, Figure 1 sketches the intuition of why different task formats may benefit from different embedding spaces.  For example, the embedding space for classification (left-hand pane) returns perfect classification performance for a linear classifier (separating triangles and circles perfectly).  However, that embedding space would not be as effective for a nearest-neighbor proximity task format, because there are some triangles whose nearest neighbors are circles, and vice-versa, which would create errors for a proximity task.  By contrast, the proximity embedding space (middle pane) always has nearest neighbors of the same class, so would be better for proximity tasks. Task formats are just one example of categorizing downstream tasks into meaningful groups for producing general-purpose embeddings; there could be more effective partitions which can be explored as part of future work.
>
> 2. While the usage of all the adapters together would require 4x the cost of storing a single embedding (or Nx depending upon number of adapters trained), we believe that storage is not a bottleneck today, being considerably cheaper than both GPU/CPU training and inference. Further, the storage cost of general-purpose embeddings would be considerably lesser compared to per-task embeddings which would scale with the number of required downstream tasks. Having said that, our approach does not increase the space cost for any individual given task. Given a task specification, users would be expected to store and generate (on demand if need be) only the actual relevant embeddings for that particular task.

---

### Official Review · Reviewer_Tbar · 2023-08-05

**Typos Grammar Style And Presentation Improvements:** none
**Soundness:** 4

**Excitement:**

3: Ambivalent: It has merits (e.g., it reports state-of-the-art results, the idea is nice), but there are key weaknesses (e.g., it describes incremental work), and it can significantly benefit from another round of revision. However, I won't object to accepting it if my co-reviewers champion it.

**Missing References:**

None

**Paper Topic And Main Contributions:**

The manuscript introduces the SciRepEval benchmark, including the 25 challenging and realistic tasks across four formats. It also provides insights into how state-of-the-art models perform on these tasks and the limitations of existing approaches. The paper also introduces a new approach that can improve performance and provides access to the best SciRep models publicly for other practitioners to use and build on.

**Questions For The Authors:**

1. Where are the 25 task datasets from, especially these datasets from 'this work'?
2. Is there any rationale why learn the task format into the representation that can help the generalization?

**Reasons To Accept:**

1. The paper introduces a comprehensive benchmark for Scientific Document Representations evaluation.
2. The paper points out that current approaches cannot generalize well on the benchmark, but the proposed method with format code can do better.


**Reasons To Reject:**

Lack of novelty.

**Reproducibility:**

4: Could mostly reproduce the results, but there may be some variation because of sample variance or minor variations in their interpretation of the protocol or method.

**Reviewer Confidence:**

2: Willing to defend my evaluation, but it is fairly likely that I missed some details, didn't understand some central points, or can't be sure about the novelty of the work.

---

> ### Author Rebuttal · Authors · 2023-08-29
>
> We thank the reviewer for the helpful comments.
>
> With regards to novelty, in contrast to the existing widely used benchmarks, we do not restrict to candidate ranking tasks but expand to 3 more task types, leading to the creation of 11 new datasets. Our proposed benchmark with 25 tasks, fulfills the need for comprehensive evaluation of scientific document embeddings where it was earlier difficult to move the needle due to lack of task coverage.
> Further, we propose and compare novel modeling techniques which not only use multi-task training, but also generate different effective embeddings for the same input based simply on the selection of the appropriate task.
>
> Answers from Authors:
>
> 1. Table 1 cites the sources for the existing datasets. The 11 new datasets were collated by us using public apis and academic search engines most prominently OpenReview and Semantic Scholar which provide paper metadata like title, abstracts, citations etc. We describe more about each of our tasks in Appendix A.
>
> 2. We build our work on the foundational idea that large scale multi-task training helps improve model generalization ability (Ye et al., 2021; Sanh et al., 2021) . Regarding why to partition by task format in particular, Figure 1 sketches the intuition of why different task formats may benefit from different embedding spaces.  For example, the embedding space for classification (left-hand pane) returns perfect classification performance for a linear classifier (separating triangles and circles perfectly).  However, that embedding space would not be as effective for a nearest-neighbor proximity task format, because there are some triangles whose nearest neighbors are circles, and vice-versa, which would create errors for a proximity task.  By contrast, the proximity embedding space (middle pane) always has nearest neighbors of the same class, so would be better for proximity tasks.

---

### Official Review · Reviewer_vkEw · 2023-08-12

**Soundness:** 4

**Excitement:**

4: Strong: This paper deepens the understanding of some phenomenon or lowers the barriers to an existing research direction.

**Paper Topic And Main Contributions:**

The work presents a benchmark, SciRepEval, which can be termed as a combination of new tasks as well as existing benchmarks and datasets such as SciDocs. The benchmark consists of 25 tasks out of which 11 are new. These tasks are split across 4 different formats, namely, classification, regression, ranking and search. The paper also presents a new approach called Multi-Format embedding, where instead of using a single embedding across the whole scientific document, they utilize different embedding based on different formats.

The paper clearly outlines the area of focus to be the following:
* If existing documents representations generalize to a highly diverse range of tasks

* If multi-task training on diverse data can improve documentation representation models

* If task-format-specific embedding representation can improve generalizability

The paper also clearly outlines the area of contributions:
* They present a new comprehensive as well as diverse benchmark for document representation techniques across 4 different formats

* An extensive investigation of generalizability of state-of-the-art scientific document representation models

* A set of new multi-task document representation models which can produce task-format specific representations. This showcases boost in generalizability and outperforms prior works by 2 points absolute



**Reasons To Accept:**

* The paper is very well written, clearly explaining the area of focus and contributions as well as providing in-depth explanation and relevant information/metadata for the steps described.

* The work presents strong reason and need for a diverse benchmark for scientific document representation, and takes a step towards that goal by providing a highly diverse benchmark.

* The paper also provides a new format-based embedding approach which showcases better performance in case of generalizability as compared to other state-of-the-art models.

**Reasons To Reject:**

* The benchmark, although diverse and better than existing benchmarks in various areas as discussed in the paper, can be termed as an incremental work at best and does not have much novelty


**Reproducibility:**

4: Could mostly reproduce the results, but there may be some variation because of sample variance or minor variations in their interpretation of the protocol or method.

**Reviewer Confidence:**

4: Quite sure. I tried to check the important points carefully. It's unlikely, though conceivable, that I missed something that should affect my ratings.

**Typos Grammar Style And Presentation Improvements:**

On page 5 on line 383 there is a typing error, maybe the authors intended to write "... and we learn _**model's capability**_ of producing an embedding for any given ..."

---

> ### Author Rebuttal · Authors · 2023-08-29
>
> Thank you for the helpful comments!
>
> We agree that SciRepEval is an incremental improvement, but we feel it is a necessary one and appropriate for the “Resources and Evaluation” track of EMNLP. Unlike existing benchmarks, we do not restrict the evaluation to candidate ranking tasks but expand to 3 more task types, leading to the creation of 11 new datasets. Our proposed benchmark with 25 tasks, fulfills the need for comprehensive evaluation of scientific document embeddings where it was earlier difficult to move the needle due to lack of task coverage.
>
>
> We would like to cite contemporary general embedding benchmarks MTEB (Muennighoff et al., EACL 2023) and BEIR (Thakur et al., NeurIPS 2021) as similar works that introduce a large collection of datasets as a standard benchmark for consistent evaluation across models. We try to fulfill the need for the same in the scientific document embedding space. We also show that our approach of learning separate embedding spaces for different task formats (nearest-neighbor, classification, regression) is effective, which is novel to our knowledge.
>
> We thank the reviewer for pointing out the grammatical mistake. We will fix it in the final version.

---

### Meta-Review · Area_Chair_pR97 · 2023-09-10

**Recommendation:** 5

**Metareview:**

This paper presents a comprehensive benchmark for Scientific Document Representations evaluation. It comprises  25 tasks out of which 11 are new. A detailed analysis of the generalizability of state-of-the-art scientific document representation models is presented along with a new method called Multi-Format embedding, where the authors  utilize different embedding based on different formats which shows better performance as compared to other state-of-the-art models.

---

### Decision · Program_Chairs · 2023-10-07

**Decision:**

Accept-Main

**Comment:**

This paper presents a comprehensive benchmark for Scientific Document Representations evaluation. It comprises  25 tasks out of which 11 are new. A detailed analysis of the generalizability of state-of-the-art scientific document representation models is presented along with a new method called Multi-Format embedding, where the authors  utilize different embedding based on different formats which shows better performance as compared to other state-of-the-art models.